# Spatiotemporal dynamics of human high gamma discriminate naturalistic behavioral states

**Abdulwahab Alasfour**[1,2]*, **Paolo Gabriel**[2], **Xi Jiang**[3], **Isaac Shamie**[3], **Lucia Melloni**[4], **Thomas Thesen**[4,5], **Patricia Dugan**[4], **Daniel Friedman**[4], **Werner Doyle**[4], **Orin Devinsky**[4], **David Gonda**[3,6], **Shifteh Sattar**[3,6], **Sonya Wang**[6,7], **Eric Halgren**[3], **Vikash Gilja**[2]

**1** Department of Electrical Engineering, Kuwait University, Kuwait City, Kuwait, **2** Department of Electrical and Computer Engineering, UC San Diego, San Diego, California, United States of America, **3** Department of Neurosciences, UC San Diego, San Diego, California, United States of America, **4** Comprehensive Epilepsy Center, Department of Neurology, New York University Grossman School of Medicine, New York City, New York, United States of America, **5** Department of Biomedical Sciences, College of Medicine, University of Houston, Houston, Texas, United States of America, **6** Rady Children's Hospital San Diego, San Diego, California, United States of America, **7** Department of Neurology, University of Minnesota Medical School, Minneapolis, Minnesota, United States of America

* abdulwahab.alasfour@ku.edu.kw

**Data Availability Statement:** Data is available at https://doi.org/10.6084/m9.figshare.16910488 and https://doi.org/10.6084/m9.figshare.20195120.

## Abstract

In analyzing the neural correlates of naturalistic and unstructured behaviors, features of neural activity that are ignored in a trial-based experimental paradigm can be more fully studied and investigated. Here, we analyze neural activity from two patients using electro-corticography (ECoG) and stereo-electroencephalography (sEEG) recordings, and reveal that multiple neural signal characteristics exist that discriminate between unstructured and naturalistic behavioral states such as "engaging in dialogue" and "using electronics". Using the high gamma amplitude as an estimate of neuronal firing rate, we demonstrate that behavioral states in a naturalistic setting are discriminable based on long-term mean shifts, variance shifts, and differences in the specific neural activity's covariance structure. Both the rapid and slow changes in high gamma band activity separate unstructured behavioral states. We also use Gaussian process factor analysis (GPFA) to show the existence of salient spatiotemporal features with variable smoothness in time. Further, we demonstrate that both temporally smooth and stochastic spatiotemporal activity can be used to differentiate unstructured behavioral states. This is the first attempt to elucidate how different neural signal features contain information about behavioral states collected outside the conventional experimental paradigm.

## Author summary

Systems neuroscience research generally relies on the experimental trial-based paradigm to reveal how the brain works. While this methodology has proved feasible and fruitful for decades, there is a need to move toward studies that leverage unstructured and naturalistic

The code to generate figures can be found in
https://github.com/WahabTNEL/PLOSCompBio.git.

**Funding:** This work has been supported, in part, by numerous UC San Diego programs. VG was the recipient of grants from the UCSD ECE Department's Medical Devices & Systems Initiative, the UCSD Centers for Human Brain Activity Mapping (CHBAM) and Brain Activity Mapping (CBAM), the Frontiers of Innovation Scholars Program, the Qualcomm Institute's Calit2 Strategic Research Opportunities (CSRO) program, the Hellman Fellowship, the Altman Clinical and Translational Research Institute, and the UCSD Office of Research Affairs Center Launch Program. PG received a grant from the Institute of Engineering in Medicine Graduate Student Fellowship. The funders had no role in study design, data collection, analysis, decision to publish, or preparation of the manuscript.

**Competing interests:** The authors have declared that no competing interests exist.

behaviors to reveal the statistical and dynamic structure of neural activity when it is not constrained to a controlled experimental setting. Here we employ a data-driven approach that shows how various signal features of high gamma band activity recorded from electrocorticography (ECoG) and stereo-electroencephalography (sEEG) can differentiate naturalistic behavioral states. These signal features include both static and dynamic aspects of the spatiotemporal neural activity. Dynamic spatiotemporal patterns extracted from high gamma band activity span multiple time scales, have a global-brain spatial representation, and better fit the data in comparison to non-dynamic approaches. These patterns individually and collectively contain valuable information differentiating between naturalistic behavioral states. This work shows that neural activity in a naturalistic setting has multiple axes of variability that must be taken into consideration in the study of the neural basis of unstructured behaviors.

## Introduction

Historically, systems neuroscience studies have relied upon trial-based accounts of behavior and stimulus-driven responses. This gold standard fails to investigate the complex and rich neural activity that occurs in a natural, unconstrained environment. Improved recording, computing, and data storage capabilities in human neuroscience allow large-scale datasets spanning days rather than minutes. Thus, we can now investigate the neural correlates of unstructured and naturalistic behavioral states and deduce the statistical structure and sources of discriminability between different states.

In a trial-based experimental paradigm, a patient or subject is instructed to complete a set of actions, engage in cognitive tasks, or passively react to external stimuli. This approach has advanced scientists' understanding of how the brain controls different external behaviors such as motor activity [1, 2] and language [3–6], or internal states such as thirst, learning and sensorimotor strategies [7–10]. The trial-based experimental approach in neuroscience serves as a proxy for understanding the brain in its natural state due to its tractable nature. However, these neural correlates may not be directly equivalent to spontaneously occurring behaviors or states that are not manifested from a structured, trial-based experimental paradigm. Continuous and naturalistic sensory-cognitive-motor loops are not only tractable but should be thought of as the main framework in which behavior, perception, and cognition should be studied [11]. Increased computational power, storage, and the ability to collect large scale multimodal data spanning days rather than minutes fosters longer-term brain studies without external constraints.

The brain has varying temporal dynamics that could span tens of seconds to hours [12], far exceeding the timespans investigated in a typical experimental setting. Band-limited power estimates of the local field potential displays fluctuations at many timescales, with the highest power present in the very low frequencies (<0.1 Hz) [13]. This slowly fluctuating band-limited power estimate is correlated across hemispheres [14], and is directly related to information accumulation [15]. Furthermore, differences in temporal timescales of neural activity are directly related to cortical hierarchy, with lower and higher-order areas having faster and slower timescales, respectively [16]. These slower timescales could also be related to brain state, and influenced by slow-activating neuromodulators that potently affect behavior [17]. Previous research usign mice models has shown that spontaneous behaviors exhibit multidimensional, brainwide activity [18]. Therefore, we assert that collecting long-term unstructured

neural activity could help researchers better understand both the spatial and temporal time scales that are relevant to describe the variability present in naturalistic behavior.

The neural correlates to naturalistic behaviors such as numerical processing [19] and arm movements [20–22] display behaviorally correlated neural activity. The ability to decode unstructured and naturalistic internal states such as mood has also been established [23]. Additionally, in a previous study, we decoded abstract unstructured behavioral states from neural activity [24].

Here, we leverage hand-labeled data collected from patients in the epilepsy monitoring unit (EMU) at an academic medical center and a pediatric medical center over the course of several days to better understand the spatial distribution and temporal timescales relevant to describing abstract behavioral states. Behavioral state labels such as "engaging in dialogue", and "using electronics," are paired with simultaneously recorded electrocorticography (ECoG) or stereo-electroencephalography (sEEG) signals. ECoG captures electrical activity directly across a significant portion of the cortical surface, while sEEG utilizes depth electrodes that record from deeper brain structures as well as the cortical surface. Our previous work [24] demonstrated the potential to identify behavioral states using conventional neural signal features used in trial-based studies. Here, we show that multiple neural signal features contribute to the neural variability present in unstructured behavior. Customary signal processing pipelines driven from trial based studies might not be ideal for capturing the full scope of neural variability outside the experimental protocol. Further, in analyzing unstructured naturalistic behavior, the lack of distinctive trial structure can preclude the ability to time-lock to specific behavioral events. We consider a problem formulation that considers statistical changes in neural activity relative to a coarse definition of behavior. In artificially inducing a trial structure to analyze behavior, an experimenter could bias their focus on behaviors arising within the confines of a trial-structure, inadvertently ignoring spontaneous activity and internal states.

To illustrate how different sources of information could influence the separability of neural activity features, Fig 1A shows a toy example of two epochs spanning 60 seconds of two behavioral states. In this illustration, we display only two simulated electrodes for simplicity. The signal we are interested in is a latent state that manifests differently on both electrodes, given behavioral state. In this example, both states are separable by multiple signal components such as channel mean, variance, covariance, and temporal dynamics. In this example, State 1 has a lower channel mean but higher variance than State 2. The covariance of both electrodes is positive in State 1 but negative in State 2. Finally, both electrodes covary slowly together in State 1, but much faster in State 2. Fig 1B displays the electrode values as a scatter plot to illustrate the differences in mean, variance, and covariance between the two states. It is imperative to investigate all of these components in the neural signal to fully understand the sources of information that discriminate between behavioral or internal states. We demonstrated that all of these components differentiate abstract and unstructured behavioral context states.

## Methods

### Ethics statement

The studies done on subjects investigated in this work were approved by the Internal Review Board of New York University (NYU) Langone Comprehensive Epilepsy Center and Rady Children's Hospital, San Diego (RCHSD) Pediadtric Epilepsy Center.

### Summary

Neural activity was collected from ECoG and sEEG electrodes from 3 subjects in the epilepsy monitoring unit. One subject was removed from further analysis due to an extensive noise in

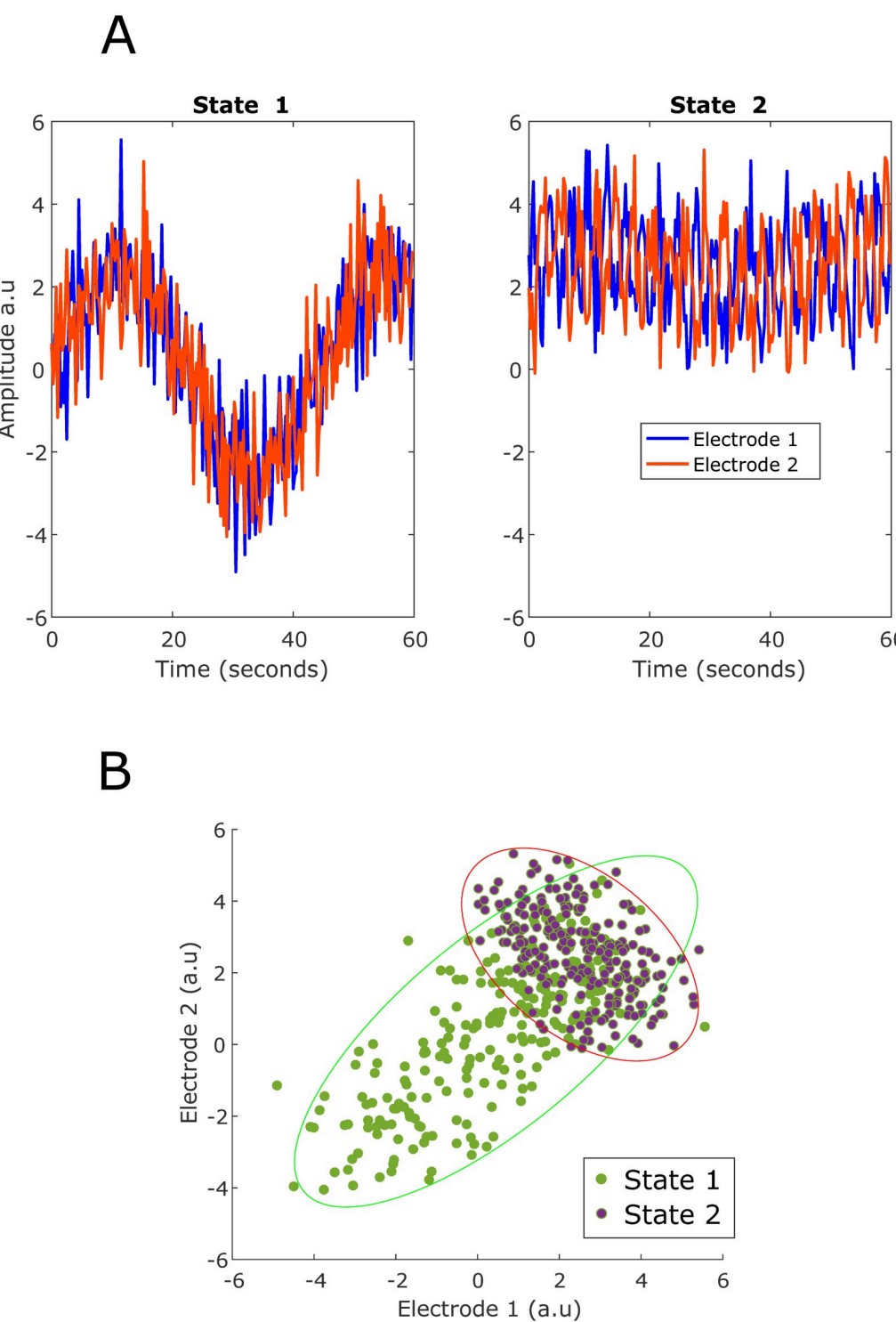

**Fig 1. Simulated electrodes visualized for two different behavioral states.** (A) Two simulated electrodes are shown to visualize different multi-variate signal components that could differentiate between two behavioral states. There is a difference between the electrodes' mean and variance when comparing states 1 and 2. Two spatiotemporal features are present at different rates in both states. There is a slow component where the electrodes are positively correlated and a fast component where the electrodes are negatively correlated. In state 1, the slow component is more prominent, while in state 2 the fast component is more prominent. (B) The values of the electrodes across both states are shown in a scatter plot. This method of visualizing electrodes could provide valuable details about the differences between the two states spatially, but it lacks any information regarding the temporal dynamics at play.

the gathered data. Subjects were not instructed to perform any specific tasks and naturalistic behavior was manually annotated using synchronized video and audio. High gamma activity was extracted from the raw neural activity by noise filtering, high-pass filtering, and Hilbert transform application, and then averaging across 250 millisecond (ms) bins. Appropriate referencing schemes were applied for the ECoG and sEEG electrodes. First, we tested whether the channels' long-term high gamma activity mean or variance could be used to decode coarse behavioral states using a support vector machine (SVM) classifier. The mean and variance were determined by averaging the high gamma activity mean or variance across 30 seconds. We next investigated whether the covariance structure of the high gamma band activity could be used for decoding. To determine whether specific temporal dynamics are relevant to naturalistic behavior, we filtered the 250 ms binned high gamma band activity using both a low pass and a high pass filter with a 1/3 Hz cut off rate. The resulting features were designated as either the "fast" or "slow" high gamma activity, and were used as inputs in an SVM classifier.

A natural extension from investigating spatial and temporal activity separately is to model the neural data's spatiotemporal activity in a unified sense. Therefore, we used Gaussian process factor analysis (GPFA) to determine whether specific spatiotemporal features in neural activity discriminate between naturalistic behavioral contexts across 30 second epochs. First, the features generated using the Expectation-Maximization (EM) algorithm were compared to features extracted from traditional factor analysis to determine if the algorithm better models naturalistic neural activity. We then applied a constrained quadratic discriminate classifier to the GPFA features to elucidate whether specific spatiotemporal activity in the high gamma band discriminates between naturalistic behavioral states. Appropriate cross-validation paradigms were applied accordingly.

## Description of data

**Subjects.**    Three subjects with intractable epilepsy with hospital ID codes corresponding to NY394, RCH1, and RCH3 participated in this study. For simplicity, they will be respectively referred as Subjects 1–3 for the remainder of this work. All subjects underwent invasive monitoring to localize epileptogenic zones before surgical resection. During their stays at New York University (NYU) Langone Comprehensive Epilepsy Center (Subject 1) and Rady Children's Hospital, San Diego (RCHSD) Pediatric Epilepsy Center (Subjects 2 and 3), the subjects, or their adult guardians for the two pediatric cases, gave their written informed consent and were subsequently enrolled in the study. A Microsoft Kinect v2 was used to record both audio and video for each subject's hospital stay using multiple modalities such as RGB, IR, and depth [25]. However, only audio and RGB video were used for this study. The audio channel was used to synchronize the video and neural streams.

**Recording.**    Subject 1 was implanted in the subdural space with platinum electrode arrays (Ad-Tech Medical Instrument Corporation, Oak Creek, WI, USA) using a combination of linear arrays consisting of 4–10 electrode contacts and a single 8 x 8 contact grid. More than 100 clinical subdural electrode contacts were embedded in SILASTIC sheets (2.3 mm exposed diameter, 10 mm center-to-center spacing) [26] and implanted directly on the right cortical surface across multiple brain regions as shown in [24]. EEG activity was recorded in the frequency range from 0.1 to 230 Hz using Nicolet clinical amplifiers, digitized and sampled at 512 Hz, and referenced to a two contact electrode array with electrodes screwed into the skull at a one cm distance from the craniotomy edge under the scalp. For this study, we focused on recordings from the 8x8 grid, given its robust coverage of frontal and temporal lobes. These cortical regions of interest have been shown to be engaged in motor [27] and language [28] related behaviors, respectively. Subjects 2 and 3 were implanted with stereo-

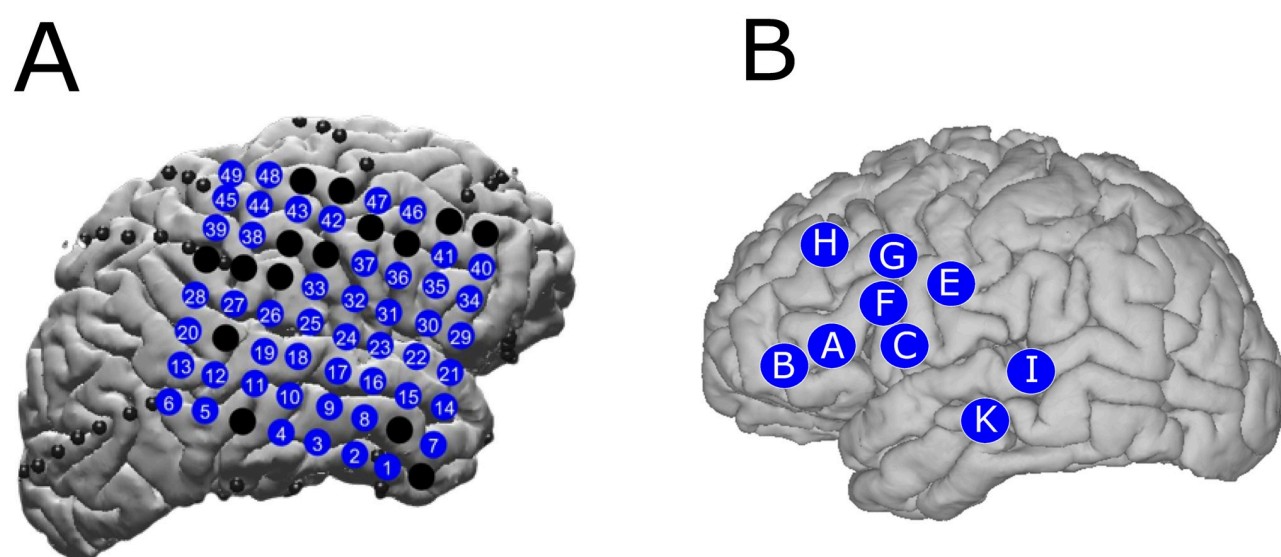

**Fig 2. ECoG grid and sEEG shanks locations.** (A) shows the coverage of the ECoG grid on the cortex for Subject 1 as well as the corresponding channel numbers. (B) shows the location of where the sEEG shanks penetrated the cortex for Subject 2 as well as the shank alphabetical labels.

electroencephalography (sEEG) electrodes via the ROSA robotic surgical implantation system (Zimmer Biomet, Warsaw, IN, USA) in various orientations to allow for intracranial recording from lateral, intermediate, and/or deep cortical and subcortical structures in a 3D arrangement [29]. sEEG electrode contacts had a 0.8 mm diameter, were 2 mm long, and were spaced 1–1.5 mm apart. Electrodes were sampled at 2000 Hz using Xltek 128Fs clinical amplifiers. For Subject 2, the sEEG implants were placed on the left frontal and temporal lobes, while for Subject 3, the implants were placed on both the left and right frontal and temporal lobes. Electrode placement for all subjects was based on clinical considerations for the identification of seizure foci. The anatomical location of the ECoG electrodes and sEEG shanks are shown in Fig 2.

## Neural signal conditioning

ECoG and sEEG data were conditioned to extract an estimate of high gamma activity (defined in this work as 70–110 Hz) across 250 ms bins while reducing the impact of noise artifacts and potential seizure activity. Prior to analysis, ECoG channels that were over seizure foci were down-selected manually by a clinician based upon visual inspection for Subject 1, while the sEEG channels for Subjects 2 and 3 did not undergo manual down-selection. sEEG channels were then referenced in a pair-wise manner with respect to neighboring electrodes. Furthermore, we have rejected sEEG channels that were outside the cortical surface. Additional ECoG and sEEG channels were rejected if their mean of the squared signal value exceeded a threshold of 3 or 1.5 standard deviations (for ECoG and sEEG respectively) above the mean squared signal value across all channels, indicative of potential noise corruption. Channel values that were equal to zero were also rejected. sEEG channels were then down-sampled from 2000 Hz to 500 Hz for faster processing (after anti-aliasing filtering). The remaining channels were then notch-filtered with center frequencies at 60, 120, 180, and 240 Hz to remove line noise and its harmonics. Finally, ECoG channels were re-referenced to a common average of all channels to remove shared noise.

Spectral features from the 70–110 Hz band of the ECoG signal were extracted by band-passing each channel. Previous studies of intracranial recordings influence our selection of this frequency band, as the high frequency activity in local field potential is heavily correlated to local

neuronal spiking activity [2, 30–32]. Additionally, the high gamma band-limited power has demonstrated power modulation with respect to sensory-motor behavior [33, 34].

The signal envelope was then calculated by taking the amplitude of the analytic signal of the Hilbert Transform. The resulting envelopes were then binned every 250 ms as previous studies have showed that this is an appropriate temporal resolution to distinguish between motor movement [1, 35], and visual stimuli [36]. Additionally, practical considerations have been taken since increasing the temporal resolution would introduce noise and increase computation time when running further analyses. The average signal amplitude was then calculated as an estimate of the square root of the power in each bin. For each continuous segment of neural activity and each channel, we removed timepoints where the value was seven times above the channel median and replaced that value using linear interpolation. We visually inspected all channels after the above preprocessing steps and rejected channels that had excess noise or variance.

Finally, we investigated whether the power spectral densities (PSD) of the band-limited amplitude (which is an estimate of the square root of the power) had a 1/f fall off, as has been shown in [12–14]. The power spectral densities of the band-limited amplitude for Subject 3's recorded data exhibited heightened activity in the high frequency components. As this is atypical relative to the previous literature, the heightened activity recorded by Subject 3's sEEG implanets may represent either noise artifacts or pathological activity. In the 49 channels that passed the channel rejection criteria and visual inspection of the time series, 32 had heightened activity in the high frequency component of the binned band-limited power. Ten examplar channels displaying this phenomenon are shown in S1 Fig. Thus, we removed Subject 3 from subsequent analyses.

## Labeling of behavioral states

Naturalistic behaviors defined "engaging in dialogue," and "using electronics" were annotated and labeled for each 5 minute segment to use as labels in our subsequent analyses. For each subject, we analyzed simultaneously recorded neural signals and clinical video/audio recording began 1–3 days (post-implant) of the subjects' stay in the hospital. Hours of continuous recordings were divided into 5 min blocks and manually labeled according to the predominant behavior observed during each video segment. A behavioral rubric was designed for this study, which categorized waking periods according to whether or not the subject was active and, if so, who or what they were interacting with. For this analysis, a subset of behavior labels were selected to represent one inactive ("Rest") and three active ("Dialogue", "Electronics", "Television") behaviors. Electronics use included using the phone, remote, tablet or playing video games. Additionally, when a subject spoke to someone on the phone, that behavior was labeled as "Dialogue", not "Electronics". Also, the "Television" label did not include the times when the subject was using the remote control.

We used three active states and one "Rest" state in our context decoding scheme for Subject 1, but only two active states ("Dialogue", "Electronics") and one rest state for Subject 2. This is due to the lack of the "Television" context when labeling Subject 2's behaviors throughout the day as Subject 2 was mainly using a tablet for entertainment purposes. See Table 1 for the number of minutes for each behavioral state for Subjects 1 and 2. See our previous work [24] for more details on the behavioral labels and their distribution across days.

## Classification of slowly varying mean and variance

The mean and variance of the high gamma band activity of the intracranial EEG channels across 30-second bins were used as features in an SVM classifier. Previous studies have shown

**Table 1. Labeling criteria and number of minutes for Subjects 1 and 2.**

| Label | Description (subject is...) | Mins S1 | Mins S2 |
|---|---|---|---|
| Dialogue | engaging in dialogue (including when the subject is talking on the phone) | 275 | 75 |
| Electronics | using an electronics device (phone, remote, tablet, video games, etc) | 40 | 130 |
| Television | watching television (does not include using remote or watching video on a tablet or phone) | 90 | N/A |
| Rest | awake but resting, either with eyes open or closed, and either with or without visitors in the room | 470 | 165 |

that fluctuations of high-gamma power that is less than 0.1 Hz is related to information accumulation [15] and inter-hemispheric correlations [14]. This was done to investigate whether long-term mean shifts or changes in high gamma band activity variance are correlated to naturalistic behavioral states. A four-class SVM with labels "Dialogue", "Rest," "Electronics" and "TV", and a three-class SVM with the above labels minus "Television" were used for Subjects 1 and 2, respectively. The multi-class SVM uses a one-versus-one coding design and a linear kernel. SVM classification was implemented using the MATLAB machine learning toolbox. We applied a 7-fold cross-validation on the data, where the test and training folds were each taken from continuous time segments rather than from randomized points across the entire dataset. The training and testing sets are class-balanced to ensure not learning a model that is biased towards the most representative class and for easier comparison to chance performance. We additionally applied a 1-fold buffer between the training and test set. This was done to reduce the potential impact of temporally local correlations between the testing and training data. Testing and training data were at least 5 mins apart. For this and subsequent analyses, we determined chance performance using the binomial cumulative distribution since we have a finite amount of data [37].

## Classification due to changes in covariance structure

The covariance matrix across 30-second epochs were calculated to determine whether the correlation or connectivity between different intracranial EEG channels could be used as a feature to classify unstructured behavioral states. To test this, we first z-scored the data across 5-min segments to remove any class-specific differences in long-term channel variances. Then, for each 30-second segment, we calculated the covariance matrix. We then applied 7-fold cross-validation with a 1-fold buffer and used a minimum distance to mean (MDM) classification algorithm to classify each class's covariance matrices, as shown in [38]. To simplify, we chose the Euclidean mean and distance as our metrics in the decoder. Other mean and distance metrics did not show a significant improvement in classification accuracy. Training and test sets were class balanced, similar to the previous sections.

## Classification due to changes in power at different time scales

Temporal dynamics of high gamma band were extracted to investigate whether they contained information regarding naturalistic behavioral states. Two time scales were considered; a slow timescale in which the temporal dynamics were slower than 3 secs and a fast timescale in which the temporal dynamics were faster than 3 secs. Mainly, we filtered the 250 ms binned high gamma band amplitude estimate using both a low-pass and a high-pass filter with a cut off frequency of 0.333 Hz (3-second period). This cutoff frequency was chosen as an extension of previous work on the data where 3-seconds was used as the temporal resolution for spectral features to classify the coarse behavioral states [24]. We then generated the signal envelope of

**Table 2. Table summarizing features analyzed.** This table summarizes how features were extracted from high gamma band activity binned in 250 ms intervals. Z-scoring is applied on the high gamma band activity first.

| Feature | Z-scoring | Preprocessing | Classification |
|---------|-----------|---------------|----------------|
| Long term mean | None | Average mean of 30 s epochs | SVM |
| Long term variance | None | Average variance of 30 s epochs | SVM |
| Covariance | 5 minute intervals | Covariance matrix of 30 s epochs | MDM |
| Fast dynamics | 5 minute intervals | High pass filter + Hilbert Transform + Average mean of 30 s epochs | SVM |
| Slow dynamics | 5 minute intervals | Low pass filter + Hilbert Transform + Average mean of 30 s epochs | SVM |
| GPFA | 5 minute intervals | Segment into 30 s epochs + GPFA fit + vectorization | QDA |

the filtered signal and took the average across 30-second bins. Prior to filtering, we z-scored each 5-min segment to ensure that discriminability between behavioral states is due to varying temporal dynamics that were not caused by ultra slow mean and variance shifts. Cross-validation and an SVM classification paradigm were then applied to the low pass and high pass filtered features using the same scheme mentioned in the previous section. Additionally, we calculated the power spectral density of each class for each channel using Welch's method. A Hanning window was used along with a 256 point Fast Fourier Transform on each 30-sec segment with 75% overlap. The power spectral densities of top performing individual channels for both subjects are shown to highlight the differences in the temporal dynamics for different behavioral contexts are shown in S2 and S3 Figs. We also show the power spectral densities of randomly subsampled sets of electrodes for both subjects in S4 and S5 Figs. Refer to Fig 2 for the locations of the ECoG electrodes for Subject 1 and sEEG bipolar pairs for Subject 2. Table 2 summarizes the features mentioned in the above sections.

## Gaussian process factor analysis

**Motivation and description.** We utilized a generative model called Gaussian process factor analysis (GPFA) to simultaneously model the spatiotemporal dynamics of neural activity. In the previous analyses, we investigated differences in the spatial and temporal dynamics for each behavioral context. However, these methods rely on investigating the differences in the spatial and temporal domains separately, not concurrently. Therefore, it is imperative to apply a data-driven method to extract spatiotemporal patterns that are relevant for each behavioral state.

In the spatial domain, frequently used dimensionality-reduction techniques such as Principal Component Analysis (PCA) or Factor Analysis (FA) look for subspaces where different channels coactivate together. PCA looks for the subspaces that exhibit the highest variance, while FA considers individual channel noise and more effectively finds common structure across sensors. However, each point in time is assumed to be independent for both methods, and no temporal structure is determined from the data. In this section, we motivate the use of GPFA as described in [39], to find spatial factors with varying autocorrelation functions. Using GPFA, one can determine each spatial component's smoothness in a data-driven way without making many assumptions about the temporal dynamics.

In GPFA, the neural activity is assumed to be generated from set of latent factors described in the following equation,

$$y_{:,t}|x_{:,t} \quad \sim \mathcal{N}(Cx_{:,t} + d, R) \tag{1}$$

Where $y_{:,t} \in \mathbb{R}^{q \times 1}$ is the neural activity of q channels at time point t, and $x_{:,t} \in \mathbb{R}^{p \times 1}$ is the latent neural state with dimensionality p at time point t (p<q). Additionally, $C \in \mathbb{R}^{q \times p}$ is the factor

loading matrix that maps the latent neural states into the observed neural activity. $d \in \mathbb{R}^{q \times 1}$ is the channel mean and $R \in \mathbb{R}^{q \times q}$ is the individual channel variance.

In a standard factor analysis model, each latent dimension $x_{i,:}$ ($i = 1, \ldots, p$) is assumed be generated from independent and identical Gaussian distributions. Therefore, there is no correlation between each consecutive time point. The GPFA augments the standard FA model by adding the assumption that the underlying latent neural activity can be correlated across different time points. GPFA defines each latent dimension across time to be generated from a Gaussian process (GP):

$$x_{i,:} \quad \sim \mathcal{N}(0, K_i) \tag{2}$$

Where $x_{i,:} \in R^{1 \times T}$ is the activity of latent state i across time. In this GP, the covariance matrix across time has the following structure

$$K_i(t_1, t_2) \approx e^{-(t_1 - t_2)^2 / 2\tau_i^2} \tag{3}$$

The full equation approximated above is shown in [39]. The correlation between time points $t_1$ and $t_2$ are related by a decaying exponential function. The parameter tau controls how fast the exponential decays. As tau increases, adjacent time points further away are correlated together, and the latent state is smoother in time. In other words, the decaying exponential controls the autocorrelation of each latent neural state.

GPFA is fit on the data using the EM algorithm [39]. Once all the parameters, $\theta = C, d, R,$ $\tau_1, \ldots, \tau_p$, are fit to the data, the p latent neural states are evaluated by determining the posterior expectation $E[x|y]$ [39].

**Reduced GPFA.** The GPFA EM algorithm looks for latent neural states with varying temporal autocorrelation functions that best fit the data. However, multiple factors could be similar in the spatial domain but vary in their temporal dynamics. Additionally, the percentage of the variance explained by each latent neural state of the data isn't explicitly determined by the algorithm. To generate latent factors with distinct spatial structures and to understand their importance in describing the data, we applied an orthonormalization procedure on the latent neural states described in [39]. The factor loading matrix C is decomposed using the singular value decomposition such that

$$C = UDV' \tag{4}$$

where $U \in R^{q \times p}$, $V \in R^{p \times p}$, and $R \in R^{p \times p}$. U and V have orthonormal columns, and D is a diagonal matrix. As the columns of C describe the vector space in which the latent neural states move in, we can rewrite the projection of the latent neural states to the channel space using an orthonormal vector space by the following operation:

$$Cx_{:,t} = U(DV'x_{:,t}) = U(\tilde{x}_{:,t}) \tag{5}$$

A new set of orthonormal neural trajectories $\tilde{x}_{:,t}$ is generated, as the space that $\tilde{x}_{:,t}$ spans is determined by the columns of U. Additionally, the columns of U are ordered by the amount of covariance explained in the data. Therefore, one can rank the latent neural states by their importance in explaining the actual neural activity. Another advantage of orthonormalization is that the orthonormal factor loading matrix U in Eq 5 is insensitive to rotations applied to the latent dimensions $\tilde{x}_{:,t}$; this is in contrast to factor matrix C shown in Eq 1, which is unique up to an arbitrary rotation matrix.

**Prediction error.** To determine whether GPFA provides insightful information regarding the temporal dynamics of different latent neural states, we compared the goodness-of-fit of

GPFA in comparison with that of FA. If GPFA were to provide a better fit than FA, then the addition of the GP temporal smoothness constraint would result in factors that better explain the recorded neural activity. To test this, we calculated the leave-one-out root mean squared error, similar to the method described in [39], using a different number of latent states. Additionally, we implemented 7-fold cross-validation with a 1-fold buffer between the training and test sets.

In this analysis, the training set was balanced to ensure that the model would not be biased to the class with the most samples. However, we used all of the test sets in each fold to reduce the variability due to the random sampling of the test sets. We calculated the root mean squared error (RMSE) of each behavioral context separately and summed them together to generate a class-balanced RMSE. We also calculated the leave-one-out class-balanced RMSE of reduced GPFA to determine whether a smaller set of orthonormal latent factors would be a better fit for the data.

**Classification of latent neural states determined from GPFA.** The GPFA EM algorithm described in the previous section is an unsupervised method for extracting spatiotemporal patterns from the data. In this work, we sought to find spatiotemporal patterns that are salient to specific unstructured behavioral states. We thus developed a GPFA-based classification method. To achieve this, we first segmented the observed neural activity into 30-second windows of 250 ms binned high gamma amplitude estimates that were z-scored for each 5-minute segment. Z-scoring was applied to remove class-specific long term mean and variance changes that would bias the EM algorithm. The dimensionality of each epoch was q x T, where q is the number of channels and T is the number of timesteps. The epochs were ordered chronologically in time for each behavioral state and segmented into K-folds.

In the training phase, we extracted a balanced number of samples for each behavioral state in the training folds to not bias the EM algorithm to look for spatiotemporal patterns specific to a certain class. We added a 1-fold buffer between the training and testing folds to minimize the effects of long term correlations. The EM algorithm was fitted on the data to extract the GPFA parameters $\theta = C, d, R, \tau_1, \ldots, \tau_p$. Once the parameters were determined, the latent neural states in both the training and test sets were determined using the posterior expectation. Each epoch of neural state activity had a dimensionality p x T, where p is the number of latent states or factors.

The epochs were then vectorized to a dimensionality of (p x T) x 1. A quadratic discriminant analysis (QDA) classifier was trained on the data and tested on a balanced testing set to facilitate a straightforward comparison to chance prediction. As the dimensionality of vectorized data (p x T) was much higher than the number of samples, it was necessary to employ a parametrization of the fitted class mean and covariance. This was achieved by calculating the following class mean and covariance.

$$\mu_C = \begin{bmatrix} \mu_{1c} \\ \vdots \\ \mu_{pc} \\ \vdots \\ \mu_{1c} \\ \vdots \\ \mu_{pc} \end{bmatrix} \tag{6}$$

$$\Sigma_C = \begin{bmatrix} \Sigma_c & \cdots & \cdots & 0 \\ \vdots & \Sigma_c & 0 & \vdots \\ \vdots & 0 & \Sigma_c & \vdots \\ 0 & \cdots & \cdots & \Sigma_c \end{bmatrix} \qquad (7)$$

Here, $\mu_{ic}$ is the mean of the latent neural state i for class c. $\mu_C$ is the class mean of the vectorized epochs. The mean is assumed to be static across time. $\Sigma_c$ is the covariance of the latent neural states for class c determined from concatenating all the epochs together, forming a matrix of dimensionality p x (T*E) where E is the number of epochs for class C. The covariance $\Sigma_C$ of the vectorized epochs for class C is constructed such that the covariance of the latent neural states at each time point is assumed to be equivalent and that time points are assumed to be independent. Even though GPFA explicitly finds temporal correlations for each neural state, this independence assumption achieves a tractable covariance matrix estimate.

In each fold, we trained and tested using the slowest latent neural state and incrementally added faster states until we included all latent states and vice versa. This was done to determine whether separability is due to slow or fast states and whether there is additive information when including both. Additionally, we trained and tested using a linear discriminant analysis where the class covariances are assumed to be equal as a control. Finally, we trained and tested using one latent neural state at a time to determine how much each latent neural state contributes to behavioral state separability. We accuracy of this classifier was calculated as the average accuracy across all cross validation folds.

## Results

### Decoding performance using first and second-order statistics

Fig 3 shows the classification performance using the mean, variance, and covariance of the high gamma amplitude estimate, and the mean of the slow and fast components. The error bars of the 7-fold cross-validation are shown. Slow and fast components are oscillations in the high gamma amplitude estimate that are slower or faster than 3 seconds, respectively. For both subjects, using any of the five signal characteristics (overall mean, variance, and covariance in addition to the fast and slow components of the high-gamma envelope), we are able to achieve a classification performance well above the finite chance level (29% and 37% for Subjects 1 and 2, respectively). This is evidence that in these naturalistic behavioral contexts, the differences in the spatial and temporal dynamics of neural activity most likely resemble the example displayed in Fig 1.

To investigate the differences in temporal dynamics of the high gamma amplitude estimate, we show the power spectral densities of exemplar channels for each behavioral state for both subjects in Fig 4. First, the power spectral density decreased monotonically as a function of frequency, corroborating previous studies [12, 13]. Most of the signal variance was clustered in the slower frequencies. Therefore, applying a decoding algorithm on the unfiltered high gamma amplitude would focus on the separability shown in the high gamma envelope's slower frequency components. This is illustrated in Fig 4, where, for example, 'Dialogue' has higher power in the slower frequencies but lower power in the higher frequencies in comparison to the other behavioral states. However, multiple channels did not exhibit a monotonically decreasing power as a function in frequency, as shown in S2 and S3 Figs.

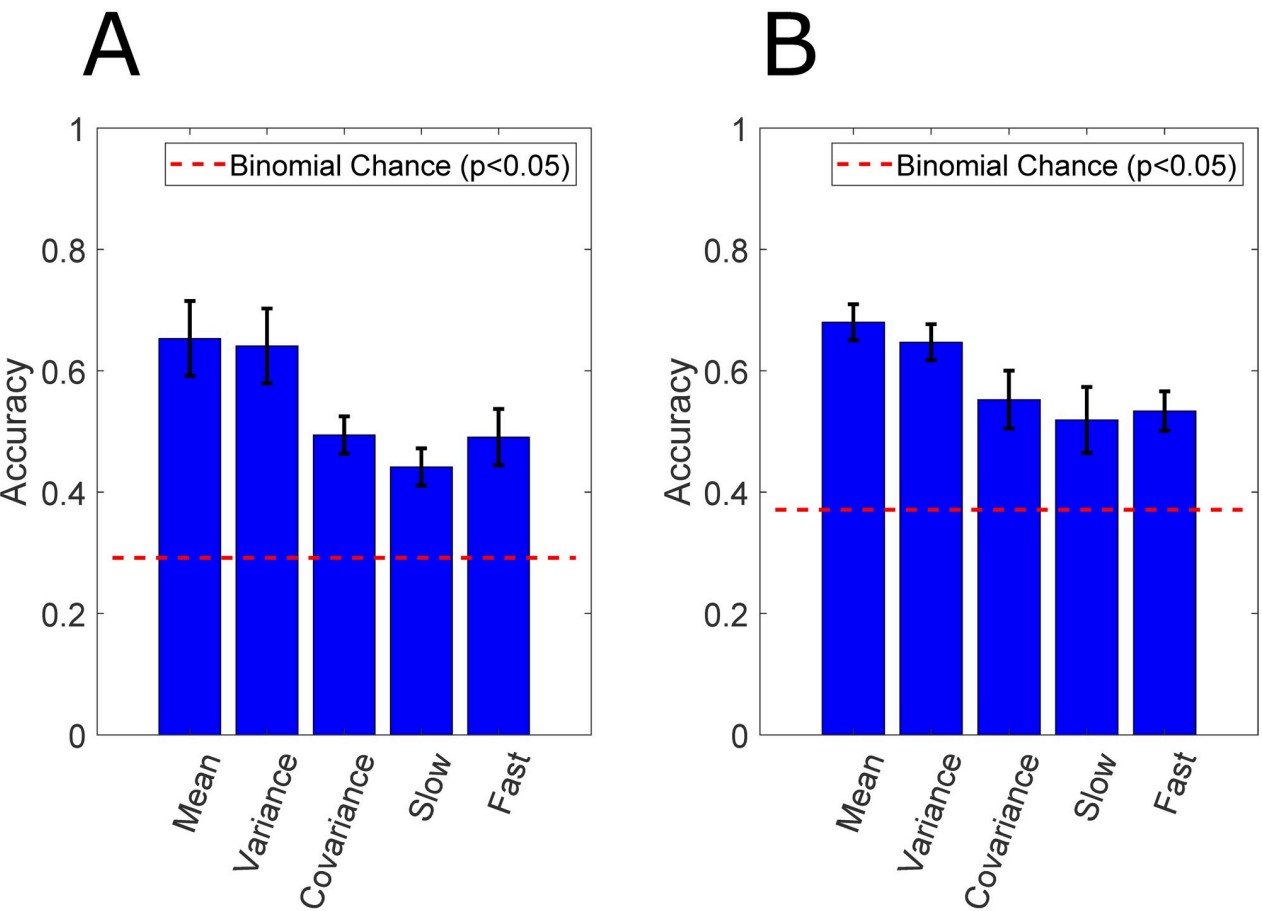

**Fig 3. Decoding performance using first and second-order statistics.** The decoding performances of Subjects 1 and 2 are shown in (A) and (B), respectively. All five different spatial and temporal components in the neural activity are able to discriminate between abstract behavioral states. The slow-varying mean and variance achieves the highest decoding performance. The covariance matrix is also able to separate behavioral states without leveraging single channel variances, as the variances are all normalized to one prior to classification. Therefore, inter-electrode correlations contribute to behavioral state separability. Finally, both the high gamma band amplitude estimate's slow and fast components contain information regarding the behavioral state.

### Reconstruction error vs. dimensionality

To assess whether the GPFA model displayed in the Methods section is an accurate representation of the data, and therefore, that the neural activity is generated from a latent subspace with variable temporal smoothness, we compared the reconstruction error vs. number of latent dimensions of FA, GPFA, and reduced GPFA. In Fig 5, we show that for both subjects and across a 7-fold cross-validation paradigm, the root mean square error (RMSE) monotonically decreases as a function of the latent space dimensionality. This shows that there exists a high dimensional subspace that could reliably describe neural activity occurring during unstructured activity. There is a clear drop in RMSE when comparing FA performance versus that of GPFA when the number of latent dimensions increases. This provides us with evidence that several spatial components in the neural activity with variable smoothness in time exist in the data, and that these spatial components, or latent factors, are repeatable and are not limited to a local temporal window.

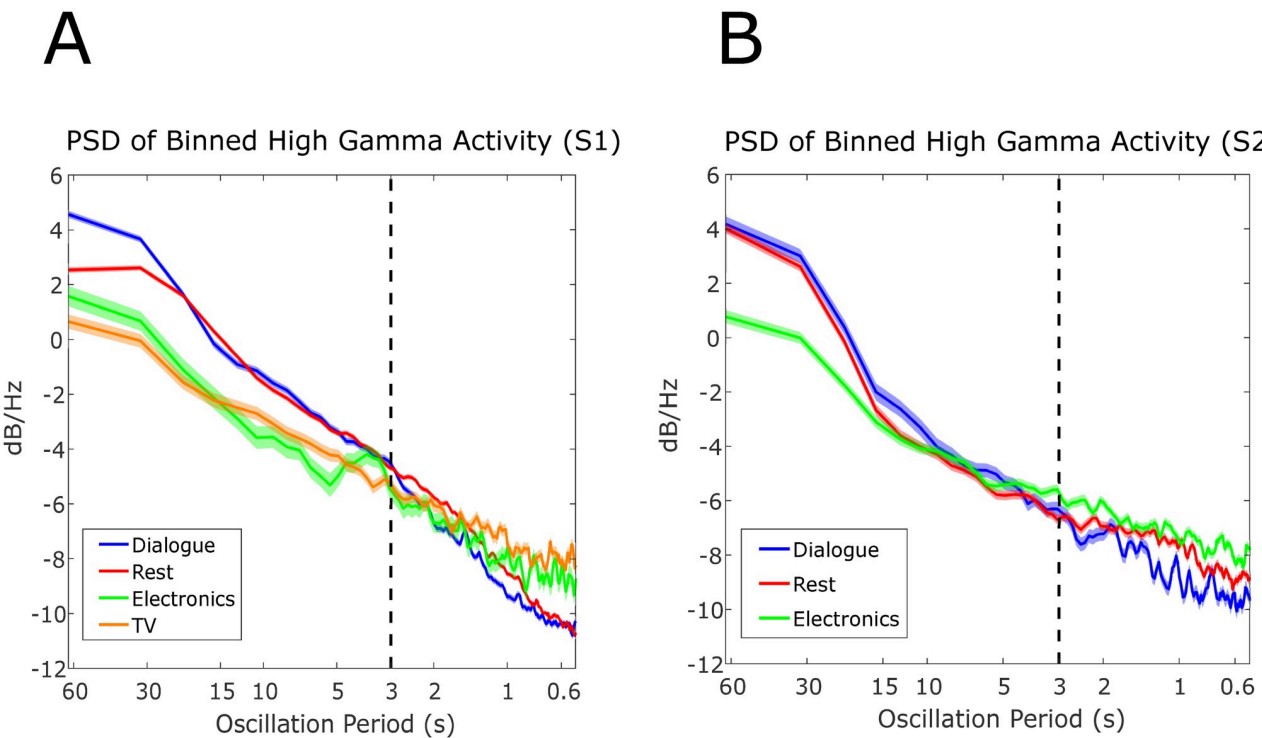

**Fig 4. Power spectral density estimates of binned high gamma amplitude estimate.** Sample electrodes are shown for Subjects 1 and 2 in (A) and (B), respectively. The power spectral densities in both sample electrodes show that different behavioral contexts exhibit varying temporal dynamics at different timescales. For example, in (A), "Dialogue" and "Rest" have more power in comparison to "Electronics" and "TV" for temporal dynamics slower than 3 seconds but vice versa for dynamics faster than 3 seconds. Most of the signal variance is concentrated in the slower components for both subjects. For Subject 1, the example electrode is channel 37 and for Subject 2 the example electrode is the bipolar pair E'9-E'10. For Subject 2, a bipolar pair with a larger number corresponds to a bipolar pair that is closer to the cortical surface. See Fig 2 for the anatomical locations of the ECoG grid and sEEG shanks. The standard error of the power spectral density estimate as calculated prior to averaging the periodograms in the power spectral density calculation. It is shown in the shaded regions.

Reduced GPFA shows that many of the spatial components found in the data are non-orthogonal after fitting 30 or 20 factors on the data for Subject 1 and 2, respectively. Fig 5 shows that there is a smaller orthonormal latent space that could reconstruct the data as well as a larger non-orthogonal latent space. We also compare GPFA and Factor Analysis's performances with varying degrees of smoothness applied to the data before fitting. This is done as a control to test whether there is a single autocorrelation function common across all latent neural states and that there are latent neural states with varying temporal dynamics. We found that there does not exist any universal autocorrelation function that led to Factor Analysis having the same or better reconstruction error as GPFA. This control provides further evidence that both slow and fast varying latent factors are repeatable in the data.

### Decoding performance using gaussian process factors

Once we have established that latent dimensions with variable temporal smoothness exist in the neural activity, a natural extension determines whether these latent dimensions contain information regarding unstructured behavioral states. Fig 6 displays the decoding performance using a QDA classifier and 7-fold cross-validation with a 1-fold buffer. We sorted the factors from slowest to fastest and vice versa to test whether behavioral state-specific information is present at either extreme. The decoder's performance increases monotonically for both

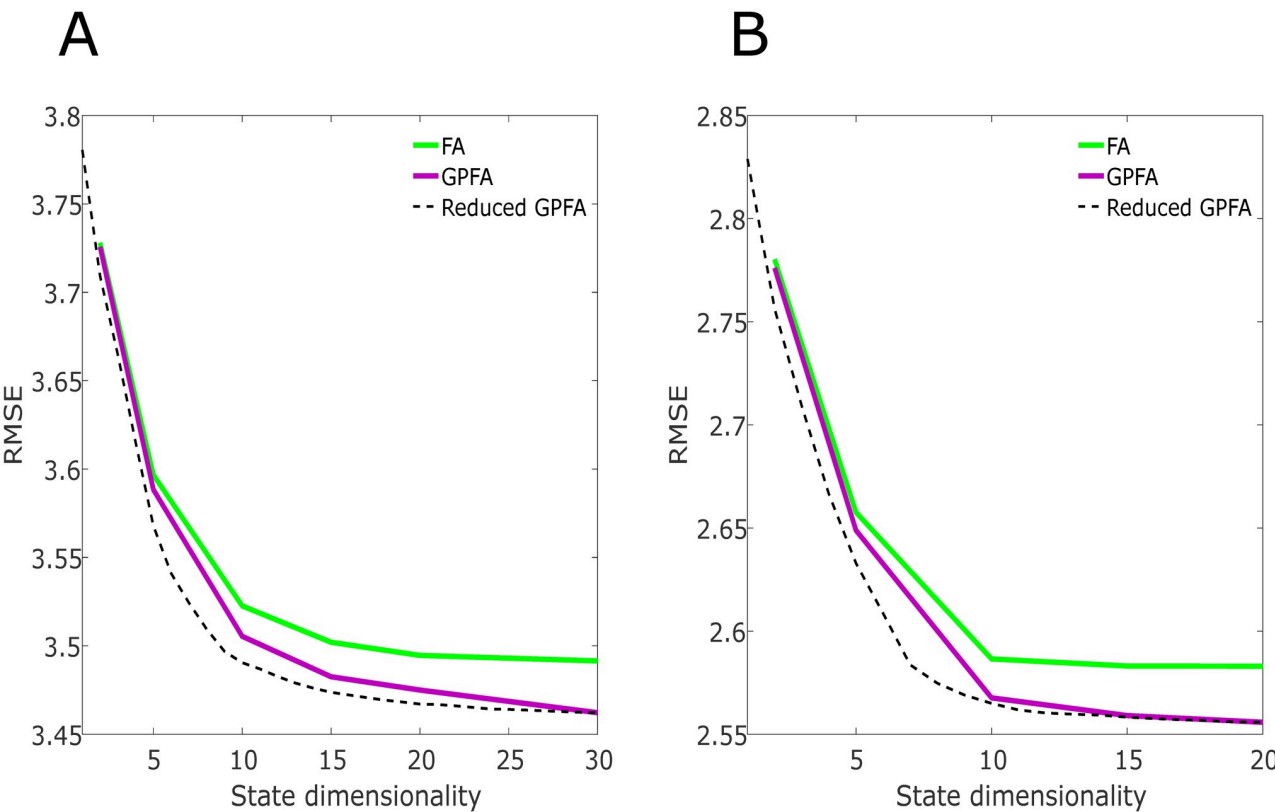

**Fig 5. Root mean square error vs. latent state dimensionality determined from FA, GPFA, and reduced GPFA.** (A) and (B) show the RMSE using the leave-one-electrode-out method for Subjects 1 and 2, respectively. For both subjects, the error decays as a function of state dimensionality, indicating the presence of complex spatial components in the neural activity. Additionally, for both subjects, the RMSE in FA is higher than GPFA for all dimensions, and the difference in RMSE increases as more latent states are learned. The orthogonal factors determined using reduced GPFA further reduces the error and shows that there exists a smaller set of spatial components that could reconstruct the neural activity as well as its higher dimension non-orthogonal counterparts.

subjects, whether starting from the slowest factor or the fastest. Using all of the factors, the decoding performances are 71% and 61% for Subject 1 and Subject 2 respectively. This shows that there is relevant information at both scales regarding behavioral state and confirms the results we described earlier. A key difference from our earlier analyses of the first and second-order statistics is that these fast and slow neural activity aspects are synchronized between multiple channels. Additionally, as a control, we applied a linear discriminant analysis (LDA) classifier to test whether the separability is due to a behavioral state-specific shift in the latent factors' mean. If that was the case, the separability of behavioral states could be due to a general mean shift in the high gamma band activity, which would manifest as shift in the latent factors' means. The LDA classifiers' performance is close to chance, suggesting that the behavioral states' separability is due to differences in the variance and covariance of the latent Gaussian process factors. In other words, the power of each factor is what is separating behavioral states.

The importance of single latent factors for decoding spanning different temporal scales is shown in Fig 7. We use each latent factor as a single feature in the QDA. Across each fold, the factors extracted from the training set might be different, therefore we show, on average, how factors at different timescales contribute to decoding performance. Only a subset of slow and fast factors contain information regarding behavioral state that is above finite chance performance. This points to the possibility that there are brain states independent of the behavioral

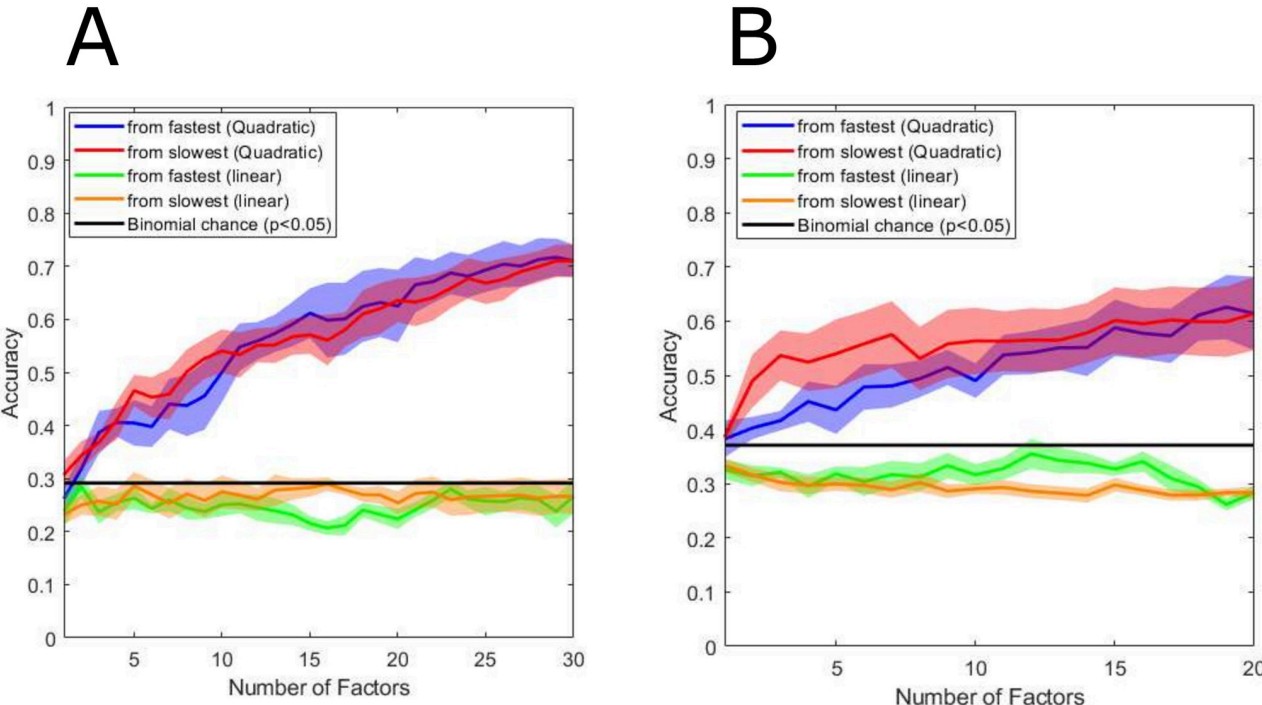

**Fig 6. Number of latent factors vs. decoding accuracy.** The decoding accuracy as a function of the number of factors generated by GPFA used (by incremental addition from slowest to fastest factors and vice versa) is shown in (A) and (B) for Subjects 1 and 2, respectively. Using a quadratic discriminant analysis as the classification algorithm, the separability of 30-second epochs increases as a function of the number of factors used. Additionally, for both subjects, there is additive information in both fast and slow factors. As a control, we used an linear discriminant analysis (LDA) to show that separability is due to differences in the latent neural trajectories' power rather than their mean (which would indicate a behavioral state-specific shift in variance or mean in the data). Standard errors of the decoding accuracies across folds are shown in the shaded region. Finite chance performance is also shown in the black line.

states we have labeled in this dataset. Additionally, no set of factors within a constrained temporal scale contributes to most of the information regarding behavioral states. This highlights that decoding performance of naturalistic behaviors using GPFA features are driven by their combinations and interactions, and as also shown in Fig 6, both the slow factors and fast factors contribute to decoding performance. Fig 7 shows the average decoding performance of the slowest to fastest spatiotemporal factors averages across folds.

## Visualization of latent factors

Fig 8 shows the latent factor weights for Subjects 1 and 2 in subsections A and B, respectively. These factor weights were generated from fitting the GPFA algorithm on the entire data-set (no crossvalidation) that is subsampled to balance the amount of data points in each class. The latent factors determined from GPFA are sorted from slowest to fastest. It is clear from Fig 8A and 8B that these factors span multiple regions in the brain, indicative that multidimensional brain-wide activity is activated during unstructured behavioral states. Furthermore, many factors show both positive and negative values for different electrodes, indicating that synchronized activation and deactivation of different regions of the brain are descriptive of naturalistic and unstructured behavioral states.

Fig 9 displays a sample epoch where the fastest and slowest factor fitted from the GPFA algorithm is shown. We display only the projections for Subject 1, as the ECoG cortical

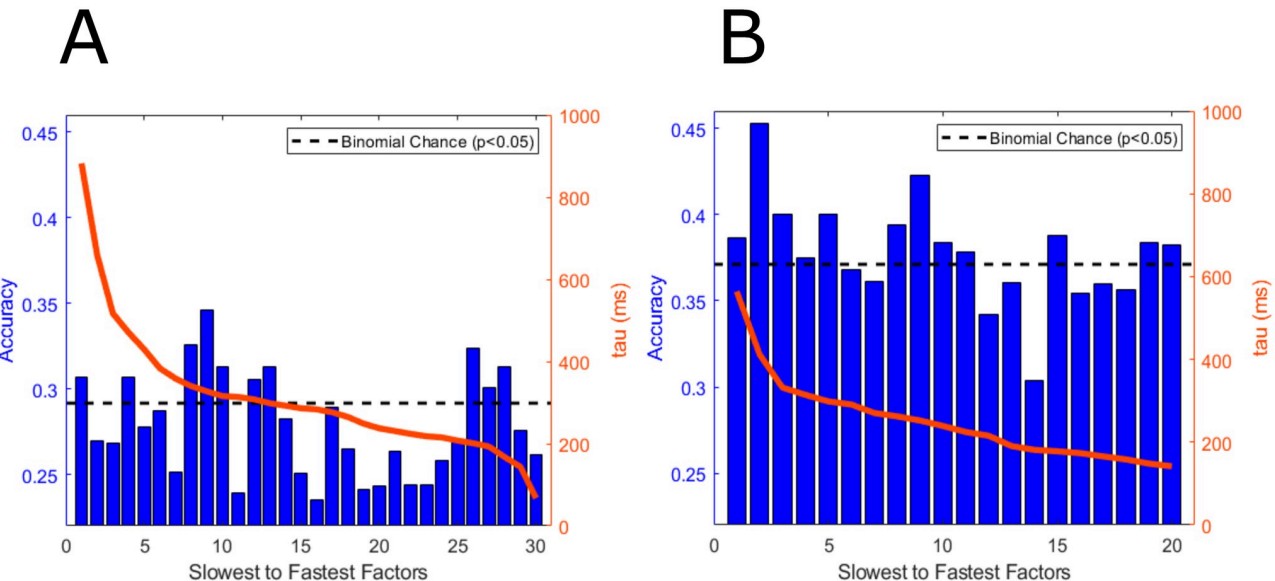

**Fig 7. Decoding accuracy of single factors.** The decoding accuracy of single factors generated by the GPFA algorithm is shown for Subjects 1 and 2 in (A) and (B), respectively. The factors are sorted from slowest to fastest. The parameter tau is fitted using the EM algorithm for GPFA and is expressed in milliseconds. We show that there is relevant information in latent factors spanning different timescales. Additionally, several factors are extracted from the data that do not contain any information regarding behavioral states. The accuracies shown are the averages of the single factor accuracies sorted from slowest to fastest across the cross validation folds. Finite chance performance is also shown in the black line.

coverage is easier to visualize than that of sEEG. For these two sample latent factors, most of the electrode projection values are all positive or all negative; therefore, the sign does not provide any additional information about the spatial structure of these factors. We multiply the fastest factor values by -1 for visualization purposes (in order to observe the synchronization between the single channel activity and the latent factor).

Fig 9B and 9D show that the fastest latent factors track multiple high frequency spikes in neural activity, while the slowest latent factor tracks the slow varying component. This visualization illustrates the existence of coherent and informative spatiotemporal patterns of unstructured behavioral states. It is important to note that these two latent factors are displayed here to show the value in using GPFA to visualize factors with different timescales. These two latent factors need not be the most important features in separating behavioral states. Additionally, these latent states are learned from a downsampled training set to ensure an even distribution of behavioral context, and therefore, do not exploit the full power of the data.

Finally, we show in S8 Fig an alternative approach in visualizing the latent factors, in which we applied the singular value decomposition algorithm to orthonormalize the latent factors and rank them according to the amount of variance explained (as shown in [39]). We can also visualize the PSDs of these orthonormal factors to understand how each orthonormal factor's temporal dynamics differ according to behavioral state, as seen in S6 and S7 Figs.

## Discussion

This paper shows that multiple sources of information in the neural activity could be leveraged to discriminate between behavioral states labeled in an unstructured setting. As demonstrated here, the mean and variance of 30 secs of the high gamma band amplitude could be used to achieve a decoding performance significantly higher than the chance level. This separability could be due to a myriad of factors such as slow shifts in brain states or slight changes in

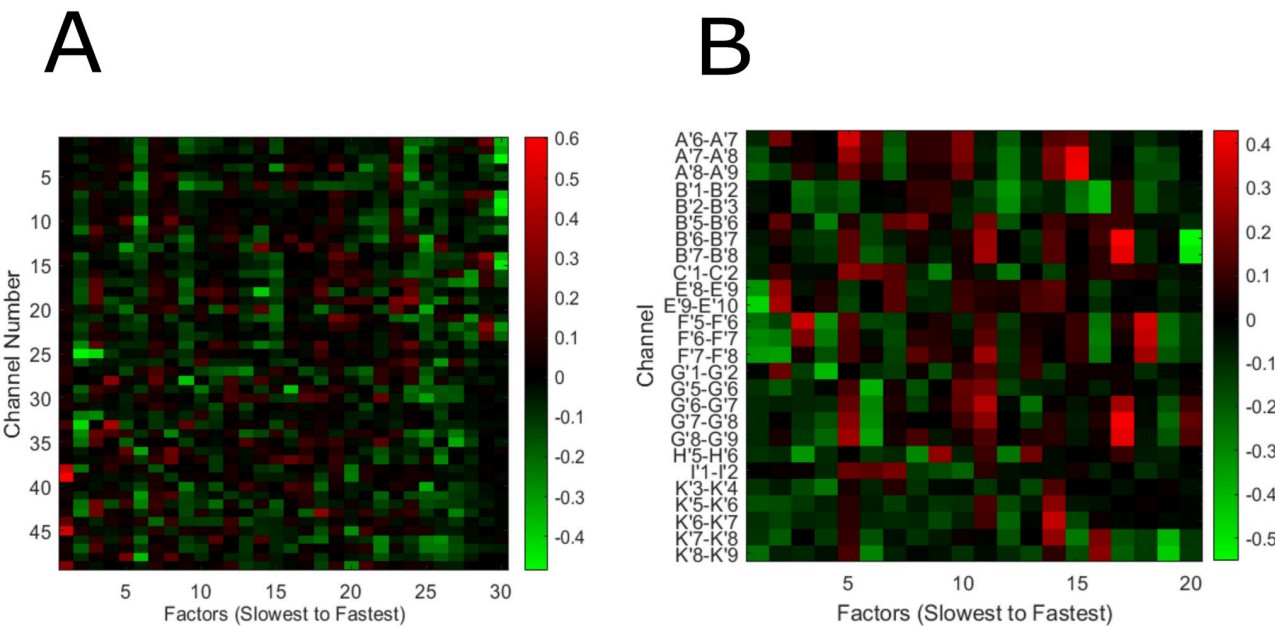

**Fig 8. Visualizations of latent factors extracted from GPFA.** We show the values of the factor weights extracted from GPFA for Subject 1 and 2 in (A) and (B), respectively. The latent factors' values show that there is synchronized activity corresponding to different areas of the brain. Additionally, some factors have both positive and negative values assigned to different channels, indicating that some factors exhibit synchronized activation and deactivation of certain brain regions. Refer to Fig 2 for the anatomical locations of the ECoG electrodes for Subject 1 and the sEEG shanks for Subject 2. Note that a bipolar pair with a larger number is closer to the cortical surface.

recording quality along with behavioral state-specific neural activity. The ability to discriminate states using the covariance matrix of the high gamma band amplitude after normalization means that decoding is still possible without any knowledge about the power in the neural activity. This confirms that the connectivity between different brain regions could be used to understand the differences between behavioral states in a naturalistic setting.

The filtered and normalized high gamma band amplitude produces both slow and fast features and provides information regarding behavioral state. Even though the decoding performance using the filtered and normalized high gamma band amplitude is less than that of using the unfiltered and unnormalized mean and variance, this result shows that both the long-term fluctuations of neural activity and faster responses separate behavioral states. The slower responses could be attributed to information accumulation of slow fluctuations in brain state, such as changes in hunger or attentiveness, driving behavior. Faster responses could be due to the brain's reaction to external stimuli.

Our results from the GPFA analyses confirm that naturalistic and unstructured behavior could be used to extract relevant spatiotemporal patterns in the neural activity. Regions of the brain where high gamma activity moves in synchrony with a specific temporal structure provide meaningful information regarding behavioral state. These spatiotemporal patterns are extracted in the absence of long term mean and variance shifts; therefore, we are confident that these patterns are not caused by some noise related brain-wide change in activity. If we applied GPFA on unnormalized neural data, an alternative scenario could be that one context has a predominantly high activity for most spatiotemporal patterns, therefore, understanding how these patterns relate to behavior would not be straightforward.

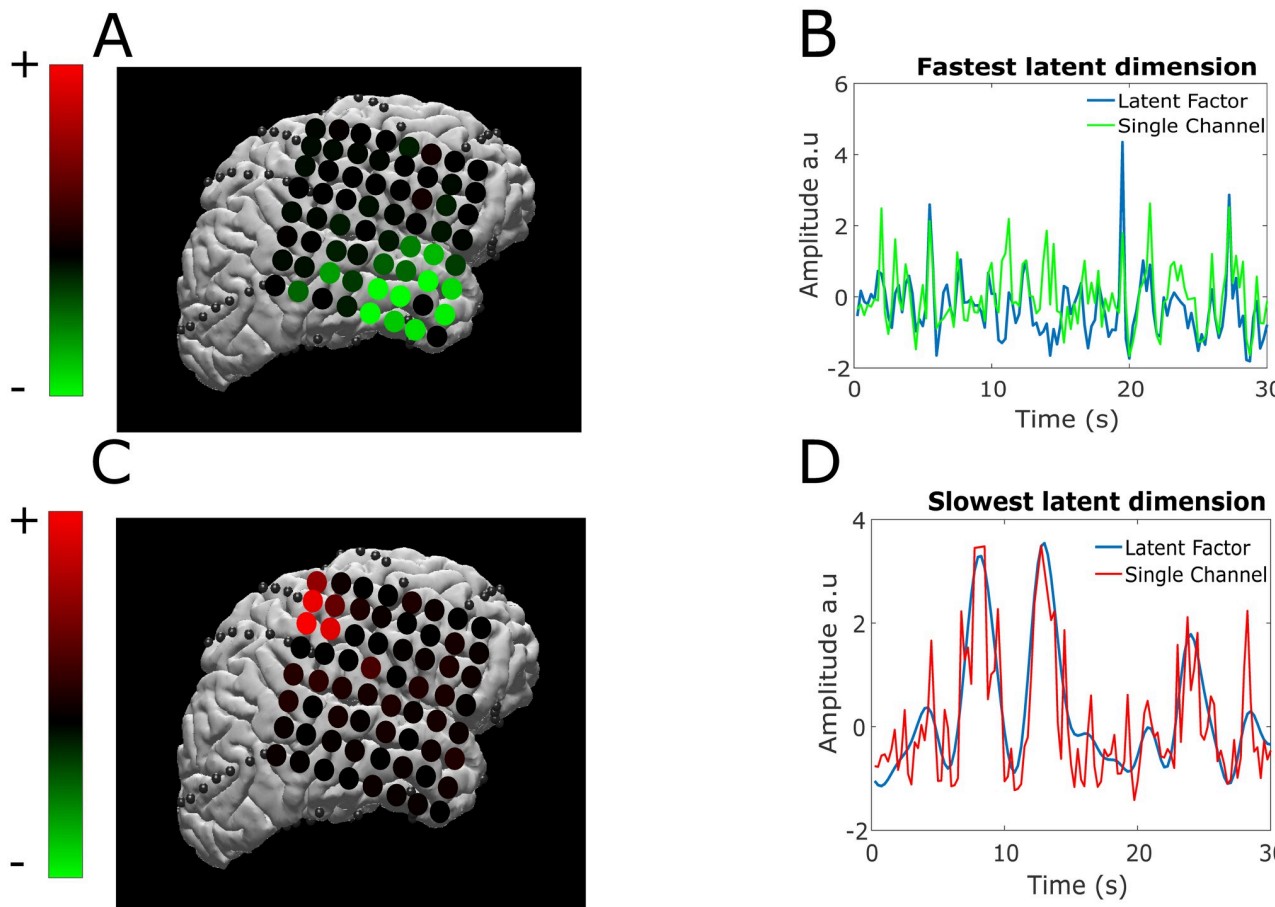

**Fig 9. Sample latent neural trajectories with variable temporal dynamics.** For Subject 1, we show the fastest and slowest latent factors extracted from the data. The factors' spatial weights are shown in the cortex in (A) and (C). Sample trajectories of the fastest and slowest factor of a single epoch are shown in (B) and (D) overlaid with the channel's value that contributes most to the corresponding factor. The factor in (B) is inverted in order to match the sign of the channel for visualization purposes. In (B), the latent factors track many high frequency peaks in the single channel amplitude. Additionally, (D) shows that the slow latent factor tracks the slow moving component of the channel amplitude.

Earlier novel algorithms have been shown to extract spatiotemporal patterns in neural activity [9, 40, 41]. However, these algorithms do not consider the many sources of variability present in long-term neural activity collected in an unstructured setting. Additionally, the presence of non-behavior-related neural activity and the lack of a trial structure in this study make this avenue of research challenging. Not only that, but relevant neural activity need not be related to externally observed changes in behavior, but instead could be due to unobserved internal activity such as changes in mood, hunger, thirst, or mental engagement. We leveraged GPFA due to the lax enforcement of numerous assumptions, such as sinusoidality, trial locking, or presence of sequences. However, GPFA and other algorithms used in similar contexts are unsupervised methods that fit a model to describe the neural activity. There is a need for supervised algorithms that extract behavior-specific spatiotemporal patterns that take into account the variability and noisiness of neural activity recorded in a naturalistic setting. Another important issue to solve is to develop these future algorithms to incorporate unbalanced datasets and take advantage of the entire dataset. Therefore, there would be no need to downsample the training set, as we did in this work.

Leveraging the power of deep learning methods could advance the analysis of naturalistic behavior. While many deep-learning models suffer from being a black box, interpreting the parameters of these models can infer physiologically-relevant features [42]. Using deep-learning algorithms such as latent factor analysis via dynamical systems (LFADS) could be a logical next step in analyzing the spatiotemporal dynamics of naturalistic behavior, but the trial-locking constraints associated with LFADS cannot fully assess naturalistic behaviors [43]. Therefore, both supervised and unsupervised deep learning models must account for multiple variability sources in order to truly understand how the brain behaves when unhindered by experimental constraints. Leveraging deep-learning techniques would be most valuable after the sources of variability in the neural data are understood, and such techniques should be designed with care after methodological analysis of the data.

Trial-based experiments are designed to find the neural correlates to specific behavior (i.e., to separate the signal from noise). However, with this study we found that multiple features of neural activity, spanning different time scales and spatial distributions, correlate to spontaneous behaviors. Huk and Hart [44] argue that researchers need to move away from linking neural activity to externally measured variables and that neural variability that could be attributed to noise is actually linked to internal states that might influence behavior. In our work, we provide evidence to support this claim.

Slow spatiotemporal dynamics clearly contribute to the separability of observable behavioral states in two subjects. In a controlled trial-based experimental paradigm, these slower dynamics could be ignored as noise. On the otherhand, the source of these dynamics can be further investigated. Drew et al. argue that subcortical modulatory systems could be driving slow oscillations of gamma band power, and that these modulations are behavior specific [45]. Factors learned from GPFA have brain-wide representation (Fig 8), confirming a previous study that showed that brain-wide latent states are correlated to spontaneous observable behaviors in mice [18]. Another study done in rats show the ability to distinguish between different brain states during naturalistic hide-and-seek play [46]. We show that in studying brain activity generated from an unconstrained, spontaneous setting across hours and days, multiple spatiotemporal features at different time scales correlate to behavior in humans. A recently published dataset by Peterson et al. [47] provides ECoG data along with coarse behavioral context labels for 12 human participants, therefore, a natural progression is to investigate whether the results obtained in this paper generalizes to more subjects and data. Different methods to quantify naturalistic behavior, as shown by McCullough and Goodhill [48], should be leveraged to better quantify the neural correlates of unconstrained behavior. We aim to inspire researchers to better understand the brain representation of behavior and internal states, and to supplement the traditional trial-based experimental paradigm with unconstrained longitudinal analyses of brain activity.

## Supporting information

**S1 Fig. Power spectral density of 10 exemplar channels for Subject 3.**
(TIF)

**S2 Fig. Power spectral density of top 10 performing channels using slow and fast temporal dynamics as features for Subject 1.**
(TIF)

**S3 Fig. Power spectral density of top 10 performing channels using slow and fast temporal dynamics as features for Subject 2.**
(TIF)

**S4 Fig. Power spectral density of randomly selected channels using slow and fast temporal dynamics as features for Subject 1.**
(TIF)

**S5 Fig. Power spectral density of randomly selected channels using slow and fast temporal dynamics as features for Subject 2.**
(TIF)

**S6 Fig. Power spectral density of top 10 orthonormal factors for Subject 1.** The percentage of variance explained for each factor is displayed in the title of each factor.
(TIF)

**S7 Fig. Power spectral density of top 10 orthonormal factors for Subject 2.** The percentage of variance explained for each factor is displayed in the title of each factor.
(TIF)

**S8 Fig. Orthonormal factor weights for both subjects.** Locations of ECoG channels and sEEG shank entry points are shown for Subject 1 and 2 in subpanels A and C respectively.
(TIF)

## Acknowledgments

We would like to thank the patients and clinicians who contributed to this study at UC San Diego, Rady Children's Hospital San Diego, and the Comprehensive Epilepsy Center at NYU Langone Medical Center. We specifically thank Preet Minas and Hugh Wang for their contributions to data acquisition, as well as Emily Bass for her help in editing the manuscript.

## Author Contributions

**Conceptualization:** Abdulwahab Alasfour.

**Data curation:** Abdulwahab Alasfour, Paolo Gabriel, Xi Jiang, Isaac Shamie, Eric Halgren, Vikash Gilja.

**Formal analysis:** Abdulwahab Alasfour.

**Funding acquisition:** Eric Halgren, Vikash Gilja.

**Investigation:** Paolo Gabriel, Lucia Melloni, Thomas Thesen, Patricia Dugan, Daniel Friedman, Werner Doyle, Orin Devinsky, David Gonda, Shifteh Sattar, Sonya Wang.

**Methodology:** Abdulwahab Alasfour.

**Project administration:** Lucia Melloni, Thomas Thesen, Sonya Wang, Vikash Gilja.

**Resources:** Vikash Gilja.

**Software:** Abdulwahab Alasfour.

**Validation:** Abdulwahab Alasfour, Vikash Gilja.

**Visualization:** Abdulwahab Alasfour.

**Writing – original draft:** Abdulwahab Alasfour, Vikash Gilja.

**Writing – review & editing:** Abdulwahab Alasfour, Paolo Gabriel, Xi Jiang, Isaac Shamie, Lucia Melloni, Thomas Thesen, Patricia Dugan, Daniel Friedman, Werner Doyle, Orin Devinsky, David Gonda, Shifteh Sattar, Sonya Wang, Eric Halgren, Vikash Gilja.

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
