## [Decision Letter · Decision Letter 0]

1 May 2022

Dear Dr. Alasfour,

Thank you very much for submitting your manuscript "Spatiotemporal Dynamics of Human High Gamma Discriminate Naturalistic Behavioral States" for consideration at PLOS Computational Biology.

As with all papers reviewed by the journal, your manuscript was reviewed by members of the editorial board and by several independent reviewers. In light of the reviews (below this email), we would like to invite the resubmission of a significantly-revised version that takes into account the reviewers' comments.

Two reviewers think your paper has definitely merits, but it has some problems that need to be fixed, in particular it needs clarification on a number of issues. I encourage you to pay close attention to these suggestions and submit a revised version.

We cannot make any decision about publication until we have seen the revised manuscript and your response to the reviewers' comments. Your revised manuscript is also likely to be sent to reviewers for further evaluation.

Sincerely,

Marieke Karlijn van Vugt, PhD

Associate Editor

PLOS Computational Biology

Daniele Marinazzo

Deputy Editor

PLOS Computational Biology

Two reviewers think your paper has definitely merits, but it has some problems that need to be fixed, in particular it needs clarification on a number of issues. I encourage you to pay close attention to these suggestions and submit a revised version.

Reviewer's Responses to Questions

**Comments to the Authors:**

Reviewer #1: This manuscript addresses the important question of how unstructured behavioral states are represented in neuronal population signals. ECoG and sEEG signals were measured from 2 human patients and high gamma band activity was extracted. The authors describe how high gamma band fluctuations at different frequencies may encode different behavioral states and the example in figure 1 clearly describes the problem. They show a novel approach for decoding bran states where slow and rapid timescales in high gamma band activity can separate behavioral states of watching TV, using electronics, engaging in dialogue and rest.

The estimation of the chance rate for the decoding performance seems to be set at .25 for subject 1 and .33 for subject 2. Simulations or the binomial formula should be used to better assess the classification results.

Previous studies have linked slow high gamma band dynamics measured in similar fashion to the fMRI signal, see for example: Drew, Duyn, Golanov and Kleinfeld, 2008, in Nature Neuroscience News and Views. It would be helpful to discuss the relation between the timescales found in the current study and this type of work.

Clarification should be provided for choosing 3 secs to separate slow from fast high gamma band dynamics.

The paper provides a nice theoretical framework for decoding brain states, but the link between different networks, timescales and factors is still a bit unclear. For example, if the factors in Figure 8 would be removed from the data, like a virtual lesion, would the same decoding still be feasible? This would help the reader understand whether a certain factor arising from a network is ‘essential’ for the approach.

Minor comments.

The scale on the y-axis differs between Figure 6A and 6B, similar axes would allow for better comparisons across subjects. It units of tau should also be stated on the axis.

There was a typo in line 180: implanets

Line 225: “The covariance matrix across 30-sec epochs were” should be matrices?

Reviewer #2: Alasfour et al. describe an analysis of naturalistic ECoG and sEEG recordings, where they show that it is possible to classify coarse behavioral states from properties of these recordings. This is an interesting dataset, and the paper is a new take on decoding behavioral states, using both engineered features and GPFA. My more specific comments are as below.

- The mean and covariance calculations were done using 30-sec or 250-ms windows. How were these parameters chosen and justified, and how would any of the results change if different decisions were made with these hyper-parameters?

- I don't find Figure 1 to be particularly helpful in the narrative of this very interesting paper. That it is possible for the covariance between multiple electrodes to change in a synthetic way that is discriminable, in simulation, is true. Perhaps this point can be better made if the real data in Fig 2 were visualized to show that it is indeed these second-order statistics in the data that are present in the data?

- How are the error shade plots in Fig 3 computed? Where in the brain are these two example electrodes? In other words, do the rest of the electrodes have the magnitude of the differences show here (beyond the biggest ones show in the supplement), and where in the brain are these electrodes located?

- I'm a little confused about the 3 subjects in this paper. Many of the results show only 2 subjects?

- How does the GPFA factor decoding compare to the first- and second-order statistics decoder? By comparing the plots, they look like they are at least in the same ballpark. Is that true quantitatively? And either way, might the authors discuss the relative merits of the different approaches to decoding?

- While I do not consider it to be a requirement for this manuscript, the authors may be interested to know about a recently published dataset, AJILE12 (Peterson et al.), which has similar ECoG data and some behavioral labels (e.g. talking, watching TV/computer) for 12 participants. It would be interesting to see if the analyses the authors presented here would generalize to another dataset with more subjects!

**Have the authors made all data and (if applicable) computational code underlying the findings in their manuscript fully available?**

Reviewer #1: **No: **Authors plan to make data and code available after acceptance for publication.

Reviewer #2: Yes

PLOS authors have the option to publish the peer review history of their article (what does this mean?). If published, this will include your full peer review and any attached files.

Reviewer #1: No

Reviewer #2: No
---

## [Decision Letter · Decision Letter 1]

18 Jul 2022

Dear Dr. Alasfour,

We are pleased to inform you that your manuscript 'Spatiotemporal Dynamics of Human High Gamma Discriminate Naturalistic Behavioral States' has been provisionally accepted for publication in PLOS Computational Biology.

Best regards,

Marieke Karlijn van Vugt, PhD

Associate Editor

PLOS Computational Biology

Daniele Marinazzo

Deputy Editor

PLOS Computational Biology

Congratulations! Both reviewers are satisfied with your revisions, and I too think this has become a wonderful paper that is worthy of publication.

Reviewer's Responses to Questions

**Comments to the Authors:**

Reviewer #1: The authors have addressed all my comments.

Reviewer #2: I thank the authors for their thoughtful revision. I am satisfied with their responses to my comments and fully support publication of the paper.

**Have the authors made all data and (if applicable) computational code underlying the findings in their manuscript fully available?**

Reviewer #1: Yes

Reviewer #2: Yes

PLOS authors have the option to publish the peer review history of their article (what does this mean?). If published, this will include your full peer review and any attached files.

Reviewer #1: No

Reviewer #2: **Yes: **Bing Brunton

---

## [Editor Report · Acceptance letter]

3 Aug 2022

PCOMPBIOL-D-22-00322R1 

Spatiotemporal Dynamics of Human High Gamma Discriminate Naturalistic Behavioral States

Dear Dr Alasfour,

I am pleased to inform you that your manuscript has been formally accepted for publication in PLOS Computational Biology. Your manuscript is now with our production department and you will be notified of the publication date in due course.

With kind regards,

Olena Szabo
